# Physical activity engagement in Eldoret, Kenya, during COVID-19 pandemic

**Karani Magutah** [1]*, **Grace Mbuthia** [2]

**1** Moi University School of Medicine, Eldoret, Kenya, **2** Jomo Kenyatta University College of Health Sciences, Juja, Kenya

* kmagutah@gmail.com, jkarani@cartafrica.org

**Data Availability Statement:** All data are in the manuscript and/or supporting information files.

**Funding:** This research was supported by the Consortium for Advanced Research Training in Africa (CARTA). CARTA is jointly led by the African

## Abstract

The World Health Organization (WHO) recommends that individuals of all ages participate in regular physical activity (PA) for optimal health and to support with the control of multiple non-communicable diseases. In Kenya however, involvement in PA across the general population is low and there is an increase in sedentary lifestyles in both rural and urban areas. An inverse relationship exists between socioeconomic status and involvement in PA. The novel COVID-19 ushered in associated control measures to limit the spread of the virus. These measures included staying at home, social distancing, and closure of physical spaces such as gyms, public parks, sports grounds, outdoor playing areas and schools. The impact was immediate, impacting patterns and routines of PA in Kenya. The primary aim of this study was to verify if COVID-19 affected PA prevalence and patterns amongst adults in Eldoret, Kenya. The secondary aim was to ascertain if the modification in behaviour is consistent amongst individuals from different socioeconomic backgrounds. We used a cross-sectional study to examine self-reported PA data amongst 404 participants. All participants were ≥18 years and resided in Eldoret, Kenya. Data were collected using a self-administered, structured questionnaire adapted from the WHO Global Physical Activity Questionnaire (WHO GPAQ). The characteristics of participants' is summarized using descriptive statistics, and bivariate analyses for measures of associations of variables was done using Chi-squared and Fishers exact tests. Binary logistic regressions were performed to adjust for the various factors and report associations between variables. The *p*-value considered for significant differences was set at <0.05. Participants in this study had mean age of 30.2 ±9.8 years. Almost 90% of the participants were not aware of the current WHO guidelines on PA, 9% stopped PA engagement after COVID-19 was first reported in Kenya, and only 25% continued regular PA. Less than half maintained PA intensity after the advent of COVID-19, with almost half reporting a drop. Males had a drop in time taken per PA session while females maintained session lengths after COVID-19 (p = 0.03). Males preferred gym-setup or mixed-type PA while females opted for indoor (home) aerobics before and after COVID-19 (p = 0.01, p = 0.02 respectively). Compared to males, females were less likely to achieve both vigorous- and moderate-intensity PA recommendations (p<0.01 and p = 0.02 respectively). Zone of residence was associated with participation in aerobic PA (p = 0.04; 95% CI = 0.02499–0.96086) and, similarly, level of education was associated with knowledge of WHO recommendations for PA (p = 0.01; 95% CI = -1.7544 - -0.2070). A majority of

Population and Health Research Center and the University of the Witwatersrand and funded by the Carnegie Corporation of New York (Grant no. B 8606R02), Sida (Grant no. 54100113), and the DELTAS Africa Initiative (Grant no. 107768/Z/15/Z) - Recipient KM. The DELTA Africa Initiative is an independent funding scheme of the African Academy of Sciences (AAS) Alliance for the Accelerating Excellence in Science in Africa (AESA) and supported by the New Partnership for Africa's Development Planning and Coordinating Agency (NEPAD Agency) with funding from Wellcome Trust (UK) and the UK government The funders had no role in study design, data collection and analysis, decision to publish, or preparation of the manuscript.

**Competing interests:** The authors have declared that no competing interests exist.

the urban population of Eldoret, Kenya and especially those with lower level of education are unaware of WHO recommendations for PA, and 30% of them have not engaged in any form of PA for many years. The majority that report involvement in PA do not achieve the WHO recommended threshold levels of PA. The results also indicated that COVID-19 has negatively affected intensity of PA, and that there has been an increase in time spent sitting/ reclining amongst individuals in the higher socio-economic classes and specifically amongst females.

## Introduction

The World Health Organization (WHO) recommends regular physical activity (PA) across all age groups to prevent and manage non-communicable diseases and promote optimal health and wellness [1]. Physical activity data are plenty and a positive curvilinear relationship between involvement in PAs of various intensities and fitness has long been established [2]. Physical activity largely contributes to a reduction in premature mortality by controlling cardiovascular disease, various cancers, diabetes and also by promoting mental health [2, 3]. This in turn translates to better overall health for individuals. It is this knowledge that has informed the existing guidelines and recommendations and also the current pool of evidence suggesting that even different exercise regimes as one form of PA would achieve similar health benefits. According to the WHO, 150 minutes of moderate-intensity PA done weekly yields the same health benefit as 75 minutes of vigorous-intensity PA. Our recent studies found that bouts of moderate-intensity PAs that are as short as 7.5 minutes, if performed thrice daily so that their cumulative time equals the WHO recommendation, yields similar health benefits, presenting additional options of regimens of PA to choose from [1, 4, 5].

Despite these WHO recommendations or other proposed options, PA involvement in Kenya remains generally low. Sedentary lifestyles are on the rise, and high inactivity levels can be found in both rural and urban areas [6–9]. Sedentary lifestyle and physical inactivity is the fourth leading risk factor for mortality globally [1] and this is worrying because in Eldoret, Kenya, 82% of elderly inhabitants do not participate in any known form of PA [10]. The start of the novel COVID-19 pandemic immediately led to control measures that entailed individuals having to social distance and stay at home. Due to the pre-existing low involvement levels of participation in outdoor PA, there was concern that the levels of PA would decline further [10]. There have been extensive debates and discussions on how to improve outdoor PA in the past, and there was now concern that even indoor home-based PAs may also be negatively impacted [1, 4, 5, 10–12]. While PA continued to be recommended during confinement associated with COVID-19 [13], the emerging data on how COVID-19 is affecting PA suggests that individuals are further reducing their participation in PA [14]. This is despite temporary recommendations suggesting home-based PAs during the COVID-19 confinement [15]. This drop has specifically been observed in older age groups and in individuals with non-communicable diseases such as diabetes [16, 17], and comes at a time when a rise in sedentary lifestyles and inactivity are associated with the increase in non-communicable diseases in Kenya and globally [6–9, 18–21]. We currently however do not have data on the effect of COVID-19 on PA in Kenya or from the eastern African region, and therefore with the absence of data are unable to paint an accurate picture of how Kenya compares globally in relation to PA changes during the pandemic.

Participation in PA has been low across all ages and populations locally and globally and while the quest to overcome this problem is unresolved [1, 4, 5, 10–12], there is a major concern that COVID-19 may reverse overall population fitness gains in general [22, 23]. Lower education and income levels have already been shown to be associated with lower involvement in PA in both males and females of all ages, and that an inverse relationship exists between social economic status and sedentary time, compounding the concern in our set up. Individuals in the lower socio economic strata have more sitting and screen-time compared to those of the higher socio economic statuses [24–28]. Local data is minimal however, and this paucity of data from lower socio-economic countries contribute the current recommendation of need for more work to adequately demonstrate how socioeconomic status impacts PA involvement [29].

Recently, new data showed that PA regimens involving shorter periods of activity that are repeated several times a day led to improved PA adherence in individuals from all socioeconomic statuses [4, 5, 30–33]. However, the start of COVID-19 mitigation measures such as physical isolation have introduced further barriers to participation in PA.

We are aware that stay-at-home-orders may negatively affect PA involvement, but we are unsure exactly how this has impacted our population's participation in PA. Recent emerging studies have documented a decline in metabolic equivalent minutes of PA in United States of America and the United Kingdom [34–36]. Given the many adjustments in behavioural aspects of life during the pandemic, we envisage a similar drop in PA participation in Kenya. Currently, the Kenyan Ministry of Health (MOH) has instructed individuals over 58 years and those with underlying health conditions to work from home and to limit outdoor engagements. Earlier, the MOH had recommended a lockdown that entailed stay-at-home for most of the population [37].

In order to develop responsive health policies, it is important to conduct research to obtain data on the current PA participation rates one year since the first case of COVID-19 was identified in Kenya. A comprehensive set of data will explain how working / staying at home and limiting outdoor engagements have affected the general population and their participation in PA, and how this may affect lifestyle diseases for the population in the future. It is hoped that this study will provide timely evidence-based data that will facilitate interventions to mitigate a potential increase in non-communicable lifestyle diseases due to the COVID-19 measures aimed to mitigate the spread of the virus [22, 23, 34–36]. The results of this study will directly contribute to the global enquiry to develop common recommendations for PA during the pandemic that can be implemented as policy by the various stakeholders [38]. The primary aim of this study is to verify if COVID-19 has affected PA prevalence and patterns amongst adults in Eldoret, Kenya. The secondary aim is to verify if the modifications to PA is the same for the different sexes and if the modifications differ depending on individuals' socio-economic backgrounds.

## Methods

### Ethics statement

Ethical approval was granted by Moi Teaching and Referral Hospital / Moi University research ethics committee (MTRH/MU IREC), approval no. 0003800. All participants provided written informed consent for participation. Participants were assured of confidentiality and anonymity, and no identifiers were used throughout the study.

### Design

This was a cross-sectional study.

## Study population and site

Participants were adults (aged ≥18 years) residing in Eldoret town and its peri-urban area within a 10 kilometers radius from the central business district (CBD).

## Sample size

The prevalence of PA participation in Eldoret is 18% [39]. This study is not only too old but was also conducted before the COVID-19 pandemic. Thus, we assumed PA involvement prevalence of 0.5, a 5% level of precision, and a Z value of 1.96 corresponding to 95% CI, yielding a sample size of 384 participants.

## Sampling procedure

Participants' selection first entailed a random selection of the estates from where a systematic criterion was employed. For participating estates, we listed estates of Eldoret town that are within a radius of ten kilometers. From these estates, we used anecdotal allocation into 2 groups for the affluent and the less affluent estates based on the availability of basic infrastructure, housing cost and affordability. An individual's income places them into certain socio-economic status largely collapsed into higher or lower economic status, and choice of residence is associated with this. Each of the two groups contributed half of the sample. A computerized random selection of four (4) estates from each category was done. We selected participants using systematic sampling based on information on the estimated number of total homes/households in the estate obtained from the village/estate elders. We determined sampling interval by first allocating the sample of 384 into the two categories equally. Thereafter, proportionate allocation based on total household numbers per estate was done, which determined the sampling interval for each of the estates by dividing the households number by the allocated sample for that estate. From the selected households, we sampled the first respondent that we encountered and although we had a replacement criterion if there was an ineligible household where respondents were below 18 years old, this never became necessary. This is summarized in Fig 1.

## Eligibility

The inclusion criteria was male and female individuals aged ≥18 years regardless of their current or previous PA history, residing within a 10-kilometer radius from Eldoret CBD, and without medical advice to keep off PAs.

## Data collection

Four trained research assistants collected data from 15[th] March to 14[th] April 2021 in private rooms or outdoors at participants' homes. The collected data included bio-demographic characteristics and self-reported information regarding PA. Data were collected using a structured questionnaire adapted from the WHO Global Physical Activity Questionnaire (WHO GPAQ). The tool included any activity that raised heart rate during performance as classified using the WHO GPAQ generic showcards that were attached to it. Thus, PA was categorized into mild, moderate or vigorous-intensity. Participants also gave an estimate of the time they spent on each activity which yielded a cumulative weekly minutes of PA. The tool was designed to either be self-administered by participants who could read and write or to be interviewer-administered for illiterate participants.

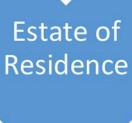

**Estate of Residence**

- List of all estates within 10 kms radius from Eldoret town central business district,
- Grouping of estates into higher (n=10) and lower (n=17) economic zones based on infrastructure and housing cost.

- All estates from both categories eligible.
- Computerized-random selection of 4 estates from each category.

**Households selection**

- 50% of sample allocated for each category.
- Determination of households number per estate in each category.

- Proportionate allocation of households for inclusion per estate
- All households with respondents ≥ 18 years old eligible

**Participants selection**

- Selection of households based on interval specific for each estate.
- Random selection of first respondent aged ≥18 years encountered.

- Systematic sampling.
- Exclusion based on medical advice and age below 18 years (n=0)
- Replacement of ineligible households (n=0) unnecessary.
- Sample (n=404) drawn, half from each category.

**Fig 1. A summary of participants' recruitment.**

## Data management and analysis

Stata version 13 (College Station, TX: StataCorp LP) was used for data entry and analysis. Univariate and bivariate analyses were performed. Continuous data such as age were summarized

**Table 1. Demographic characteristics.**

| Variable | Zone 0 (Lower income) Mean ± SD or n (%) | Zone 1 (Higher income) Mean ± SD or n (%) | Overall Mean ± SD or n (%) |
|---|---|---|---|
| Mean age (N = 404) | 30.6±10.3 (n = 202) | 29.9±9.3 (n = 202) | 30.2±9.8 (n = 404) |
| | [Males 29.0±8.5 (n = 94); Females 31.9 ±11.6 (n = 108)] | [Males 28.8±8.2 (n = 96); Females 30.8 ±10.2 (n = 106)] | [Males 28.9±8.3 (n = 190); Females 31.4 ±10.9 (n = 214)] |
| Education level (N = 400) | | | |
| None | 4 (2) | 2 (1) | 6 (1.5) |
| Primary | 50 (25) | 13 (6.5) | 63 (15.75) |
| Secondary | 85 (42.5) | 58 (29) | 143 (35.75) |
| Tertiary | 61 (30.5) | 127 (63.5) | 188 (47) |
| Employment (N = 402) | | | |
| None | 77 (38.3) | 83 (41.3) | 160 (39.8) |
| Self Employed | 87 (43.3) | 63 (31.3) | 150 (37.3) |
| Formally Employed | 37 (18.4) | 55 (27.4) | 92 (22.9) |
| Work entails physical exertion (N = 402) | | | |
| Yes | 77 (38.7) | 48 (24.9) | 125 (31.9) |
| No | 122 (61.3) | 145 (75.1) | 267 (68.1) |

using means and their standard deviations while categorical data such as PA time-length were summarized using frequencies and percentages to depict participants' characteristics based on their zones of residence described above. Bivariate analyses were based on the various PA-related variables studied per sex groups and age categorizations. These bivariate analyses for measures of associations were done using Chi-squared tests. Odds ratios and 95% confidence intervals were reported when associations were significant for binary outcomes associated with PA. To report associated factors and adjust for the various confounders, logistic regressions at 95% confidence interval were performed for effect of various independent variables on select binary outcomes. $P$ values of $<0.05$ were considered statistically significant.

## Results

The mean age of the participants was 30.2±9.8 years (males 28.9±8.3 (n = 190); females 31.4 ±10.9 (n = 214)). One in every four participants had a minimum of secondary level of education and the majority were in the working class. The demographic characteristics are shown in Table 1.

### Physical activity awareness and patterns

Almost 9 in 10 participants were unaware of the current WHO guidelines on PA and the majority (52.9%) of these respondents came from the lower-income zones. Nine percent of the participants reported that they had stopped exercising after COVID-19 was first reported in Kenya, and most of these individuals lived in lower-economic zones/estates. Based on reported session duration and number of days of PA per week, only 25% of participants achieved the recommended threshold of weekly minutes of PAs, a majority of whom came from the lower-economic estates. Physical activity awareness and reported PA history are shown in Table 2 while patterns of PA are shown in Table 3.

Bivariate analyses showed that while individuals from lower-income estates were more likely to participate alone in PA (p = 0.01), in organized gym sessions (p = <0.001) and preferred outdoor running (p = 0.02), those from higher-income estates did PA either alone or with others, preferred indoor PA at home or in the gym, and performed a mixture of PAs which included park/estate running both before and after COVID-19. Those from high socio-economic estates preferred gym aerobics and indoors/home aerobics while those from lower socio-economic estates preferred walking. Before COVID-19, the lower socio-economic estate individuals preferred doing morning PAs as opposed to the evening as preferred by the higher socio-economic estates' participants. Table 4 summarizes variables with significant differences between zones.

Males were twice more likely to be aware of the WHO recommendations for PA compared to females (OR = 2.09 (95% CI 1.05–4.30); p = 0.02). Females tended to participate in PA alone or in combination with a family member whereas males preferred to exercise in groups or at the gym (p = <0.001). Females also preferred mild-to-moderate intensity PAs while males preferred vigorous-intensity (P = <0.001). The odds of achieving the recommendations for vigorous-intensity PA were 5.21 (95% CI 2.35–12.7) times higher in males than females. The odds of achieving moderate-intensity PA recommendations were also higher in males than females (OR 1.6 (95% CI 1.06–2.46). Before COVID-19, males had longer sessions of PA (p = 0.01) and this sex difference was lost after COVID-19. Males had a drop in time taken per PA session while females maintained session lengths after COVID-19 (p = 0.03). Males also preferred gym-setup or mixed-type PAs while females opted for indoor (home) aerobics before and after COVID-19 (p = 0.01, and p = 0.02 respectively). Further, both before and after COVID-19, males preferred running and ball game PAs as opposed to walking and indoor aerobics for

**Table 2. Physical activity history and awareness.**

| Variable | Zone 0 (lower-income) n (%) | Zone 1 (higher-income) n (%) | Overall n (%) |
|---|---|---|---|
| Last date of hard labour / PA (N = 399) | n = 199* | n = 200* | |
| a) Stopped (none) since March 2020 | 21 (10.6) | 13 (6.5) | 34 (8.5) |
| b) None for years | 63 (31.7) | 79 (39.5) | 142 (35.6) |
| c) Within the last 3 days | 57 (28.6) | 43 (21.5) | 100 (25.1) |
| d) Within the week but >3 days ago | 10 (5.0) | 14 (7) | 24 (6.0) |
| e) 1–2 weeks ago | 9 (4.5) | 16 (8) | 25 (6.3) |
| f) 3 weeks—3 months ago | 20 (10.1) | 17 (8.5) | 37 (9.3) |
| g) 3–12 months ago | 19 (9.6) | 18 (9) | 37 (9.3) |
| Awareness of WHO recommendations | | | |
| (N = 398)          Yes | 17 (8.6) | 27 (13.5) | 44 (11.1) |
|                    No | 181 (91.4) | 173 (86.5) | 354 (88.9) |
| Considers self aerobically active (N = 398) | | | |
|          Yes | 145 (72.5) | 127 (64.1) | 272 (68.3) |
|          No | 55 (27.5) | 71 (35.9) | 126 (31.7) |
| Last planned PA (N = 388) | | | |
|   a) Stopped (none) since March 2020 | 13 (6.8) | 14 (7.1) | 27 (7.0) |
|   b) None for years | 61 (31.8) | 54 (27.6) | 115 (29.6) |
|   c) Within the last 3 days | 68 (35.4) | 48 (24.5) | 116 (29.9) |
|   d) Within the week but >3 days ago | 13 (6.8) | 19 (9.7) | 32 (8.3) |
|   e) 1–2 weeks ago | 8 (4.2) | 32 (16.3) | 40 (10.3) |
|   f) 3 weeks—3 months ago | 18 (9.4) | 18 (9.2) | 36 (9.3) |
|   g)>3–12 months ago | 11 (5.7) | 11 (5.6 | 22 (5.7) |

females (p<0.001 for both). Males also preferred planned (morning or evening) PAs while females participated in PA only as time became available both before and after COVID-19 (p = 0.02 and p = 0.03 respectively). Table 5 summarizes associations of sex and the various variables, with odds ratios provided for binary outcomes.

To determine the associated factors and degree of such associations, a binary logistic regression model to predict factors that affected current participation in aerobic PA while controlling for other confounders showed that only zone of residence had statistically significant association (p = 0.04; 95% CI = 0.02499–0.96086) where individuals from lower socio-economic zones had higher participation. Age, sex, knowledge of WHO recommendation for PA and level education all were not significantly associated with participation in PA. Similarly for knowledge of WHO recommendations for PA, only level of education was associated (p = 0.01; 95% CI = -1.7544 - -0.2070) such that individuals with higher education levels were more likely to be aware of existing recommendations compared to those with lower education. Variables such as zone of residence, age, and sex were not associated with knowledge of WHO recommendations for PA (all p>0.05).

## Discussion

To the best of our knowledge, this is the first study to report on how COVID-19 has affected PA amongst individuals in Kenya. The current study found that 9 in 10 participants were unaware of the current WHO guidelines that recommend 150 or 75 weekly minutes of moderate- or vigorous-intensity PA, respectively [1]. Bio-demographic characteristics of age and sex had no statistical association with this knowledge and only level of education was statistically associated where individuals with higher education were more likely to be aware of existing

**Table 3. Patterns of pas.**

| Variable | Zone 0 (Lower income) n (%) | Zone 1 (Higher income) n (%) | Overall n (%) |
|---|---|---|---|
| PA intensity (N = 272) | | | |
| Mild | 44 (30.3) | 40 (31.5) | 84 (30.9) |
| Moderate | 89 (61.4) | 70 (55.1) | 159 (58.5) |
| Vigorous | 12 (8.3) | 17 (13.4) | 29 (10.7) |
| PA intensity change since COVID-19 (N = 272) | | | |
| Maintained intensity | 57 (39.3) | 58 (45.7) | 115 (42.3) |
| Dropped intensity | 65 (44.8) | 46 (36.2) | 111 (40.8) |
| Increased intensity | 23 (15.9) | 23 (18.1) | 46 (16.9) |
| Mod. intensity attainment of 150 mins (N = 268) | | | |
| Yes | 90 (62.1) | 90 (73.2) | 180 (67.2) |
| No | 55 (37.9) | 33 (26.8) | 88 (32.8) |
| Vig. intensity attainment of 75 mins (N = 265) | | | |
| Yes | 15 (10.4) | 24 (19.8) | 39 (14.7) |
| No | 129 (89.6) | 97 (80.2) | 226 (85.3) |
| Weekly days of PA (N = 270) | | | |
| Daily | 41 (28.3) | 22 (17.6) | 63 (23.3) |
| ≥4 | 49 (33.8) | 52 (41.6) | 101 (37.4) |
| ≤3 | 33 (22.8) | 33 (26.4) | 66 (24.4) |
| Rarely | 22 (15.2) | 18 (14.4) | 40 (14.8) |
| Weekly days of PA pre-COVID-19 (N = 272) | | | |
| Daily | 43 (29.6) | 28 (22.1) | 71 (26.1) |
| ≥4 | 50 (34.5) | 55 (43.3) | 105 (38.6) |
| ≤3 | 22 (15.2) | 23 (18.1) | 45 (16.5) |
| Rarely | 30 (20.7) | 21 (16.5) | 51 (18.8) |
| Bout lengths (today) (N = 267) | | | |
| <10 mins | 14 (9.9) | 13 (10.4) | 27 (10.1) |
| ≥10 - <30mins | 47 (33.1) | 43 (34.4) | 90 (33.7) |
| ≥30 mins | 81 (57.0) | 69 (55.2) | 150 (56.2) |
| Bout lengths (pre-COVID-19) (N = 265) | | | |
| <10 mins | 22 (15.7) | 15 (12) | 37 (14.0) |
| ≥10 - <30mins | 35 (25) | 40 (32) | 75 (28.3) |
| ≥30 mins | 83 (59.3) | 70 (56) | 153 (57.7) |
| PA length change since COVID-19 (N = 265) | | | |
| Maintained bout length | 27 (19.3) | 21 (16.8) | 48 (18.1) |
| Dropped bout length | 43 (30.7) | 41 (32.8) | 84 (31.7) |
| Increased bout length | 70 (50) | 63 (50.4) | 133 (50.2) |
| Pre COVID-19 place of PA (N = 269) | | | |
| Gym | 8 (5.6) | 16 (12.7) | 24 (8.9) |
| Home (indoors) | 23 (16.1) | 21 (16.7) | 44 (16.4) |
| Outdoors (park / estate | 100 (69.9) | 66 (52.4) | 166 (61.7) |

(*Continued*)

**Table 3.** (Continued)

| Variable | Zone 0 (Lower income) n (%) | Zone 1 (Higher income) n (%) | Overall n (%) |
|---|---|---|---|
| Mixed (≥2 above) | 8 (5.6) | 19 (15.1) | 27 (10.0) |
| Others | 4 (2.8) | 4 (3.2) | 8 (3.0) |
| Current place of PA (N = 269) | | | |
| Gym | 4 (2.8) | 10 (8.0) | 14 (5.2) |
| Home (indoors) | 28 (19.4) | 38 (30.4) | 66 (24.5) |
| Outdoors (park / estate | 99 (68.8) | 63 (50.4) | 162 (60.2) |
| Mixed (≥2 above) | 8 (5.6) | 12 (9.6) | 20 (7.4) |
| Others | 5 (3.5) | 2 (1.6) | 7 (2.6) |
| Pre COVID-19 PA types (N = 267) | | | |
| Running | 44 (31.2) | 44 (34.9) | 88 (33.0) |
| Walking | 60 (42.6) | 35 (27.8) | 95 (35.6) |
| Aerobics (all types) | 29 (20.6) | 38 (30.2) | 67 (25.1) |
| Ball games | 8 (5.7) | 9 (7.1) | 17 (6.4) |
| Current PA types (N = 265) | | | |
| Running | 43 (30.5) | 40 (32.2) | 83 (31.3) |
| Walking | 62 (44.0) | 31 (25.0) | 93 (35.1) |
| Aerobics (all types) | 27 (19.1) | 42 (33.9) | 69 (26.0) |
| Ball games | 9 (6.4) | 11 (8.9) | 20 (7.6) |
| Work-out day of the week now (N = 264) | | | |
| Any day of week | 104 (75.3) | 95 (75.4) | 199 (75.4) |
| Only weekends/holidays | 19 (13.8) | 17 (13.5) | 36 (13.6) |
| Only weekdays | 15 (10.9) | 14 (11.1) | 29 (11.0) |
| Work-out day of the week pre-COVID-19 (N = 264) | | | |
| Any day of week | 110 (79.1) | 96 (76.8) | 206 (78.0) |
| Only weekends/holidays | 17 (12.2) | 13 (10.4) | 30 (11.4) |
| Only weekdays | 12 (8.6) | 16 (12.8) | 28 (10.6) |
| Preferred PA time now (N = 260) | | | |
| Any time available (doesn't matter) | 66 (48.5) | 50 (40.3) | 116 (44.6) |
| Morning between 6 and 8 am | 37 (27.2) | 27 (21.8) | 64 (24.6) |
| Evening between 5 and 7 pm | 29 (21.3) | 42 (33.9) | 71 (27.3) |
| Night (between 7pm and 6am) | 4 (2.9) | 5 (4.0) | 9 (3.5) |
| Preferred PA time pre-COVID-19 (N = 259) | | | |
| Any time available (doesn't matter) | 68 (50.4) | 55 (44.4) | 123 (47.5) |
| Morning between 6 and 8 am | 35 (25.9) | 19 (15.3) | 54 (20.9) |
| Evening between 5 and 7 pm | 28 (20.7) | 44 (35.5) | 72 (27.8) |
| Night (between 7pm and 6am) | 4 (3.0) | 6 (4.8) | 10 (3.8) |

(*Continued*)

**Table 3.** (Continued)

| Variable | Zone 0 (Lower income) n (%) | Zone 1 (Higher income) n (%) | Overall n (%) |
|---|---|---|---|
| Mean minutes spent sitting/reclining (N = 369) | 258.5±175.8 range (19.8–900). Males (n = 84) 289.3±199.8 Range (30–900) Females (n = 102) 233.1±149.6 range (19.8–720) | 311.3±194.4 range (30–1020). Males (n = 89) 335.6±189.1 Range (30–1020) Females (n = 94) 288.4±197.6 range (30–840) | 284.7±186.9 range (19.8–1020). Males; n = 173 313.1±195.2 range (30–1020) Females; n = 196 259.6±176.0 range (19.8–840) |
| Time spent sitting/reclining since COVID-19; N = 393 | | | |
| Increased | 115 (58.7) | 105 (53.3) | 220 (56.0) |
| Maintained | 67 (34.2) | 75 (38.1) | 142 (36.1) |
| Reduced | 14 (7.1) | 17 (8.6) | 31 (7.9) |

recommendations for PA. Except for level of knowledge where the current study shows an association, our findings replicate what has already been shown elsewhere [40]. Our study was carried out in a region of Kenya renown for producing world-class athletes and this could probably explain why individuals with higher level of education may be aware of the recommendation for PA as they probably read about and follow on performance of such elite athletes, a feat individuals with less education may not study through or relate.

One in three participants had not engaged in any PA for years, which mirrors WHO reports that 25–33% of individuals worldwide do not engage in PAs at all [41]. A total of 68% of participants considered themselves aerobically active, however only 30% of these individuals actually achieved the recommended threshold of weekly minutes of PA. This translated to an overall of 25% prevalence in attainment of PA recommendations. Amongst the PA achievers, the majority came from the lower-economic estates, tended to be males, and were performing moderate-intensity PAs. Zone of residence was statistically associated with participation in PA where individuals from lower-economic zones were more likely to be involved when compared to those from the higher-economic zones. The findings are consistent with data showing that lower, rather than higher income, is positively associated with higher PA involvement [41]. While our current work only examined individuals from one town in Kenya with differing socio-economic status, the findings do however mirror other studies comparing higher- and lower-income countries which show that lower-income individuals tend to be more active than their higher income counterparts [41]. There is however an ongoing debate in the literature with dissenting studies refuting this association [42].

While it has been shown that it is still possible for some to attain PA recommendations during the COVID-19 pandemic [43], 9% of participants in our current study stopped exercising after COVID-19 was first reported in Kenya, and the majority of these participants came from lower economic zones/estates. Less than half of the participants maintained previous PA intensity after the advent of COVID-19, with almost half reporting a drop in activity. It should be noted that there was no change in preferred day-of-the-week for PA, PA type, session length

**Table 4. Association between zone (reference, low socioeconomic) of residence and various variables.**

| | Pearson χ2 | P value |
|---|---|---|
| Participation in hard physical labour | 8.62 | <0.001 |
| Normalcy of exercise (alone, group, combinations) | 14.2 | 0.01 |
| Preferred exercises location (gym/home/park/estate/mix) before COVID | 20.6 | <0.001 |
| Preferred exercises location (gym/home/park/estate/mix) currently | 16.6 | <0.001 |
| Preferred exercises (running/ball games/walking/gymnastics) currently | 9.57 | 0.02 |
| Preferred time-of-day for a workout before COVID (morning) | 8.58 | 0.04 |

**Table 5. Association between sex (reference, males) and various variables.**

| | Pearson χ2 | P value | Odds Ratio (binary outcomes) |
|---|---|---|---|
| Knowledge of WHO exercise recommendation | 5.15 | 0.02 | 2.09 (1.05–4.30) |
| Participation in hard physical labour | 0.33 | 0.57 | |
| Whether currently active in aerobic exercises | 0.06 | 0.80 | |
| Normalcy of exercise (alone, group, combinations) | 15.7 | <0.001 | |
| Current exercise intensity | 22.5 | <0.001 | |
| Whether exercise intensity changed since COVID-19 | 5.08 | 0.08 | |
| Whether meeting 150 minutes moderate intensity exercise | 5.64 | 0.02 | 1.61 (1.06–2.46) |
| Whether meeting 75 minutes vigorous intensity exercise | 21.2 | <0.001 | 5.21 (2.35–12.7) |
| Regularity of aerobic exercises (current) | 0.60 | 0.90 | |
| Regularity of aerobic exercises (before COVID-19) | 2.0 | 0.57 | |
| Exercise session length | 9.12 | 0.01 | |
| Exercise session length (before COVID-19) | 5.88 | 0.05 | |
| Change in exercise session length | 7.3 | 0.03 | |
| Exercise location (gym/home/park/estate/mix) before COVID-19 | 14.9 | 0.01 | |
| Current exercises location (gym/home/park/estate/mix) | 12.0 | 0.02 | |
| Pre-COVID exercise type (running/ball games/walking/gymnastics) | 50.7 | <0.001 | |
| Preferred exercises (run/ball games/walk/gymnastics) currently | 36.9 | <0.001 | |
| Preferred day of workout before COVID-19 | 1.94 | 0.38 | |
| Preferred day of workout currently | 5.23 | 0.07 | |
| Preferred time-of-day for workout before COVID-19 | 10.3 | 0.02 | |
| Preferred time-of-day for workout currently | 8.9 | 0.03 | |
| Change in time spent sitting/reclining since COVID-19 | 0.79 | 0.68 | |

or place of PA for those who maintained their activity levels during the pandemic. The time spent sitting increased for the majority of the participants, with only one-third maintaining pre-COVID-19 sitting time and only 8% of participants reducing their sitting time, echoing studies elsewhere [44, 45]. Reviews of recent studies have highlighted that COVID-19 may have negatively impacted PA involvement as per the results of our study [46, 47]. Our study further added that the higher socio-economic strata was more adversely affected during COVID-19 as it pertain to PA levels, although there was an overall drop in PA levels across both higher and lower socioeconomic classes. Only 25% of participants, who mostly hailed from lower economic estates, reported exercising regularly. These individuals from the lower-socioeconomic estates who participated in PA reported doing PA while alone, in organized outdoor-exercises (park or estate), and preferred to do walks. This differed from those from higher socio-economic status who preferred various combinations for whom to exercise with, and who, further, preferred PAs done indoors either at home or at a gym (or mix), and preferred aerobic activities both before and after COVID-19. It appears COVID-19 did not change preferences of where to do PA or with whom for all participants. The start of the novel COVID-19 did however affect the preferred time of day for PA. Before COVID-19, participants from higher socio-economic estates preferred evening PAs and those in the lower socio-economic estates tending to morning hours. However, after COVID-19, this association was lost. This could be associated with COVID-19 mitigation measures that have reduced opportunities for PAs due to the suspension of outdoor engagement prospects. We still however have insufficient data to fully demonstrate the entire impact of the pandemic on PA [44, 47–49].

Females were twice less likely than males to be aware of the WHO recommendations for PA, and, were more likely to participate alone in PA or in combination with a family member

compared to group/gym sessions for males. Males were however more likely to have vigorous-intensity PAs compared to mild-to-moderate intensities for females and were 5 times more likely to achieve WHO recommendations for vigorous-intensity PA. They were also 1.6 times more likely to achieve moderate-intensity PA recommendations compared to their female counterparts. There however was no significant association between sex and awareness of WHO recommendations for PA or participation in the same. Previous research has shown that males reduced vigorous-intensity PA while females increased moderate-intensity PA after COVID-19 [45]. Our current work however, adds that males and females maintained their PA intensity preferences even with reported overall decline in PA involvement after COVID-19. Before COVID-19, being male was associated with longer average sessions of PA, but this ceased with the start of the pandemic resulting in a significant drop in time taken per PA session. This differed from females who managed to maintain their pre-COVID-19 session lengths. Recent research has shown that the pandemic has negatively impacted the types and intensities of PA with which individuals engage, and daily sitting time has increased [45]. Except for exercise intensity, there has not been any segregation based on sex and age that we are aware of [44, 45]. Our study attempts to segregate by sex, with additional variables. Concerning the venue for PAs, both males and females maintained their preference for gym set-up. Additionally, males seemed to maintain mixed-type PA, running and ball games while females maintained indoor (home) activities, walking and indoor aerobics. The current study also found that being male was associated with planned (either morning or evening) PA while being females was associated with PA only as time became available, without prior planning both before and after the start of COVID-19.

## Limitations

The cross-sectional design employed in the current study was unable to assess the actual effect of COVID-19 on PA participation. Our attempt to handle this may have introduced a recall bias that might have affected the data for the period before COVID-19. We attempted to reduce this by including only those questions we thought had minimal recall challenges, but we note that this may not have totally eliminated the bias.

## Conclusions

A majority of the urban population of Eldoret, Kenya and especially those with lower level of education were unaware of the WHO recommendations for PA and 30% of them have not engaged in any PA for years. For those reporting participation in PA, the majority do not achieve the recommended WHO threshold levels. COVID-19 has reduced participation in PA and increased the time spent sitting/reclining especially for individuals in higher socio-economic class and for females.

## Supporting information

**S1 Data.**
(XLSX)

## Author Contributions

**Conceptualization:** Karani Magutah.

**Data curation:** Karani Magutah.

**Formal analysis:** Karani Magutah, Grace Mbuthia.

**Funding acquisition:** Karani Magutah.

**Investigation:** Karani Magutah.

**Methodology:** Karani Magutah.

**Project administration:** Karani Magutah.

**Resources:** Karani Magutah.

**Software:** Karani Magutah.

**Supervision:** Karani Magutah.

**Validation:** Karani Magutah.

**Visualization:** Karani Magutah.

**Writing – original draft:** Karani Magutah, Grace Mbuthia.

**Writing – review & editing:** Karani Magutah, Grace Mbuthia.

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
