## [Decision Letter · Decision Letter 0]

17 Nov 2021

PGPH-D-21-00875

Exercise Involvement among urban populace of Eldoret, Kenya, after advent of COVID-19.

Dear Dr. Magutah,

Thank you for submitting your manuscript to PLOS Global Public Health. After careful consideration, we feel that it has merit but does not fully meet PLOS Global Public Health’s publication criteria as it currently stands. Therefore, we invite you to submit a revised version of the manuscript that addresses the points raised during the review process.

EDITOR: Please see the reviewers comments and improve the manuscript by addressing each of the comments.

The decision of this manuscript is justified on PLOS Global Public Health’s publication criteria and not on novelty or perceived impact.

We look forward to receiving your revised manuscript.

Kind regards,

Zulkarnain Jaafar

Academic Editor

1. We suggest you thoroughly copyedit your manuscript for language usage, spelling, and grammar. If you do not know anyone who can help you do this, you may wish to consider employing a professional scientific editing service.

2. We ask that a manuscript source file is provided at Revision. Please upload your manuscript file as a .doc, .docx, .rtf or .tex. If you are providing a .tex file, please upload it under the item type ‘LaTeX Source File’ and leave your .pdf version as the item type ‘Manuscript’.

3. Please update the completed 'Competing Interests' statement, including any COIs declared by your co-authors. If you have no competing interests to declare, please state "The authors have declared that no competing interests exist". Otherwise please declare all competing interests beginning with the statement "I have read the journal's policy and the authors of this manuscript have the following competing interests:"

4. In the online submission form, you indicated that "data is available upon reasonable request"

5. Please amend your detailed Financial Disclosure statement. This is published with the article, therefore should be completed in full sentences and contain the exact wording you wish to be published.

i). State the initials, alongside each funding source, of each author to receive each grant.

ii). State what role the funders took in the study. If the funders had no role in your study, please state: “The funders had no role in study design, data collection and analysis, decision to publish, or preparation of the manuscript.”

6. Please ensure that the funders and grant numbers match between the Financial Disclosure field and the Funding Information tab in your submission form. Note that the funders must be provided in the same order in both places as well. Currently, some funders are listed in the Financial Disclosure but not on the Funding information section.

Additional Editor Comments (if provided):

Reviewers' comments:

Reviewer's Responses to Questions

**Comments to the Author**

1. Does this manuscript meet PLOS Global Public Health’s publication criteria? Is the manuscript technically sound, and do the data support the conclusions? The manuscript must describe methodologically and ethically rigorous research with conclusions that are appropriately drawn based on the data presented.

Reviewer #1: No

Reviewer #2: Yes

2. Has the statistical analysis been performed appropriately and rigorously?

Reviewer #1: Yes

Reviewer #2: I don't know

3. Have the authors made all data underlying the findings in their manuscript fully available (please refer to the Data Availability Statement at the start of the manuscript PDF file)?

Reviewer #1: No

Reviewer #2: Yes

4. Is the manuscript presented in an intelligible fashion and written in standard English?

Reviewer #1: No

Reviewer #2: No

5. Review Comments to the Author

Reviewer #1: 

1) A flow chart should be used to show the samples recruitment adopting both inclusion and exclusion criteria at each stage.

2) I have not found a single line about ‘How physical activities were measured’? What are the domains the authors considered? How they calculated MET value? Need to describe the measurement of physical activity in a separate paragraph.

3) The authors mentioned the aim of this study was ‘to determine the prevalence and regimes of exercise amongst adults one year after the first case of COVID-19 was reported in Kenya’. I am confused about the term ‘exercise’. How the authors examined exercise using GPAQ is misleading here? The authors need to clarify precisely the difference between exercise and physical activity and how GPAQ helped them to quantify exercise?

4) Authors should describe how they developed the data collection tool and what were the variables they considered to achieve the objective of this study?

5) Authors need to describe the data collection procedure in details. Who collected the data? How privacy was maintained?

6) Why the authors divided the results into lower-income and higher-income zone? Did it align with the study objectives?

7) In Table 3, the authors reported exercise intensity: mild, moderate and vigorous. However, was it work-related or recreational or transport-related? Authors need to clarify this issue precisely.

8) Throughout the manuscript the authors mixed the term ‘exercise’ and ‘physical activity’. Kindly confirm which was actually you measured?

Reviewer #2: 

General comments: This study fills out an important gap in the literature and is providing important information. Nonetheless, my opinion is that the paper needs to be rewritten and improved in order to be published. The manuscript should also be checked for language mistakes.The following paragraphs detail what might, in my opinion, improve the quality of the paper.

Abstract

Background, 4th line: “both rural and urban areas.” Please could you specify if this is in the world or in Kenya?

Background, last line: “amongst adults.” Please specify where (in Eldoret?)

Introduction

Please provide a source for the first paragraph. I’m unsure if that information is necessary in the context of the paper. I would have preferred more details on the benefits of PA on health instead.

Why are the authors focusing on “exercises” instead of physical activity in the paper? In my opinion, physical activity would be a better concept to put forward for the paper, instead of exercises. It might be interesting to explain the difference between the two concepts. The WHO usually uses physical activity (not exercises) for its guidelines.

Please provide at least one source for “Exercise data is plenty and it has long been established that…”

The last sentence of the first paragraph of Introduction has many English mistakes.

“In Kenya, more than 1 in 4 (82%)”: confusion. Is it more than 3 in 4?

There are many repetitions in the Introduction. The structure needs to be revised to bring one main idea per paragraph. Please avoid repeating the same information in every paragraph.

Regarding the aim of the study (last sentence of the introduction), please specify where after “amongst adults” (in Eldoret?) Suggestion: I think it would improve the clarity of the paper very much if the aim of the paper was more specific and if another secondary aim was put forward. For example, the Results section presents all data according to socioeconomic zones. It might be worth it to express it as an aim for the paper (to verify if the prevalence varies according to the socioeconomic background of the population?) In the discussion, it almost sounds like the aim of the study was to verify if COVID-19 has affected physical activity in Eldoret, Kenya. Is this the case? Maybe the aim of the paper was to verify if COVID-19 has affected physical activity prevalence and patterns amongst adults in Eldoret, Kenya? And the secondary aim is to verify if the modification is the same between socioeconomic background?

Methods

In the section “Data management and analysis,” more information about how the data were analyzed is needed. For example, all the tables in the paper present the data based on 2 socioeconomic zones, but this is not explained in the method section. I would like to know why and how these zones were created. I expect the “Data management and analysis” section to explain clearly how the data were handled to answer the aim of the paper. Right now, the lack of structure makes it look like a lot of random analysis were done, without a clear purpose. Finally, I think it is also important to clearly detail when the reporting of odds ratios was applicable.

Results

In the Tables, I would like to see for which variables there are significant differences between zones. The overall % could be presented alone when there is no difference.

Many results are presented in the text without being aggregated in a Table. I agree that all results don’t need to be in a Table, but if multiple gender-based analyses have been conducted, a Table should also support these results.

The “Patterns of exercises” table provides a lot of information. I would like to be able to see where there are significant differences between pre- and since COVID.

The results that directly answer the aim of the study should be clearly stated in a paragraph. There are many variables and many analyses, and the reader gets lost. Why are the authors adding white- and blue-collar results at the end? The aim of the study could be more defined and the Method and Result sections should be more aligned with it. All the variables don’t need to be presented in the same paper.

Discussion

In the first paragraph, the authors mention that a study showed that low-income individuals are more active than their higher income counterparts. I did not find this information in the WHO fact sheets. Could the authors bring more information about this? I don’t think this is the case in high income countries (might be worth mentioning). It would be highly interesting to dig a little more on that topic, given that all the results are shown according to socioeconomic status.

The last 2 paragraphs of the discussion are results, not discussion. Please discuss your results with the literature.

6. PLOS authors have the option to publish the peer review history of their article (what does this mean?). If published, this will include your full peer review and any attached files.

**Do you want your identity to be public for this peer review?** For information about this choice, including consent withdrawal, please see our Privacy Policy.

Reviewer #1: **Yes: **Dr. Lingkan Barua

Reviewer #2: No

---

## [Decision Letter · Decision Letter 1]

8 Feb 2022

PGPH-D-21-00875R1

Physical Activity Engagement in Eldoret, Kenya, during COVID-19 Pandemic.

Dear Dr. Magutah,

Thank you for submitting your manuscript to PLOS Global Public Health. After careful consideration, we feel that it has merit but does not fully meet PLOS Global Public Health’s publication criteria as it currently stands. Therefore, we invite you to submit a revised version of the manuscript that addresses the points raised during the review process.

Dear authors,

Kindly attend to all the reviewers comments in order to improve your manuscript for a better standard of scientific reporting.

The decision of this manuscript is justified on PLOS Global Public Health’s publication criteria and not on the novelty or perceived impact of your work.

We look forward to receiving your revised manuscript.

Kind regards,

Zulkarnain Jaafar

Academic Editor

Journal Requirements:

Additional Editor Comments (if provided):

Reviewers' comments:

Reviewer's Responses to Questions

**Comments to the Author**

1. If the authors have adequately addressed your comments raised in a previous round of review and you feel that this manuscript is now acceptable for publication, you may indicate that here to bypass the “Comments to the Author” section, enter your conflict of interest statement in the “Confidential to Editor” section, and submit your "Accept" recommendation.

Reviewer #1: All comments have been addressed

Reviewer #2: All comments have been addressed

2. Does this manuscript meet PLOS Global Public Health’s publication criteria? Is the manuscript technically sound, and do the data support the conclusions? The manuscript must describe methodologically and ethically rigorous research with conclusions that are appropriately drawn based on the data presented.

Reviewer #1: Partly

Reviewer #2: Yes

3. Has the statistical analysis been performed appropriately and rigorously?

Reviewer #1: No

Reviewer #2: Yes

4. Have the authors made all data underlying the findings in their manuscript fully available (please refer to the Data Availability Statement at the start of the manuscript PDF file)?

Reviewer #1: Yes

Reviewer #2: Yes

5. Is the manuscript presented in an intelligible fashion and written in standard English?

Reviewer #1: Yes

Reviewer #2: Yes

6. Review Comments to the Author

Reviewer #1: 1) Your ‘introduction’ should describe the evidence regarding PAL among lower and high-income groups to justify your secondary objectives. I have not yet found anything regarding this.

2) You need to make the Flow chart of sample recruitments more standard. Use arrow under each stem of recruitment & beside the arrow mention the technique you adopted. If any participants excluded, show the numbers & reasons of exclusion.

3) Did you adopt binary logistic regression? What were the factors did you adjust? If not, kindly adopt binary logistic regression to report associated factors.

4) Did you apply any technique to control recall bias?

Reviewer #2: I congratulate the authors for answering all the comments. I enjoyed reading this new version of the manuscript. I still have some minor comments (see attachment)

7. PLOS authors have the option to publish the peer review history of their article (what does this mean?). If published, this will include your full peer review and any attached files.

**Do you want your identity to be public for this peer review?** For information about this choice, including consent withdrawal, please see our Privacy Policy.

Reviewer #1: No

Reviewer #2: No

---

## [Editor Report · Decision Letter 2]

22 Mar 2022

Physical Activity Engagement in Eldoret, Kenya, during COVID-19 Pandemic.

PGPH-D-21-00875R2

Dear Dr. Magutah,

We are pleased to inform you that your manuscript 'Physical Activity Engagement in Eldoret, Kenya, during COVID-19 Pandemic.' has been provisionally accepted for publication in PLOS Global Public Health.

Best regards,

Zulkarnain Jaafar

Academic Editor